# TcJAV3–TcWRKY26 Cascade Is a Missing Link in the Jasmonate-Activated Expression of Taxol Biosynthesis Gene *DBAT* in *Taxus chinensis*

**DOI:** 10.3390/ijms232113194

**Published:** 2022-10-29

**Authors:** Li Chen, Ling Wu, Liu Yang, Haiyang Yu, Pingliang Huang, Yuehua Wang, Ruifeng Yao, Meng Zhang

**Affiliations:** 1State Key Laboratory of Chemo/Biosensing and Chemometrics, Hunan Provincial Key Laboratory of Plant Functional Genomics and Developmental Regulation, College of Biology, Hunan University, Changsha 410082, China; 2Shenzhen Research Institute of Hunan University, Shenzhen 518055, China

**Keywords:** JAV, WRKY, JA signal transduction, *DBAT*, taxol biosynthesis

## Abstract

Jasmonates (JAs) are the most effective inducers for the biosynthesis of various secondary metabolites. Currently, jasmonate ZIM domain (JAZ) and its interactors, such as MYC2, constitute the main JA signal transduction cascade, and such a cascade fails to directly regulate all the taxol biosynthesis genes, especially the rate-limit gene, *DBAT*. Another JA signaling branch, JAV and WRKY, would probably fill the gap. Here, TcJAV3 was the closest VQ-motif-containing protein in *Taxus chinensis* to AtJAV1. Although TcJAV3 was overexpressed in *AtJAV1* knockdown mutant, *JAVRi17*, the enhanced disease resistance to *Botrytis cinerea* caused by silencing *AtJAV1* was completely recovered. The results indicated that TcJAV3 indeed transduced JA signal as AtJAV1. Subsequently, TcWRKY26 was screened out to physically interact with TcJAV3 by using a yeast two-hybrid system. Furthermore, bimolecular fluorescence complementation and luciferase complementary imaging also confirmed that TcJAV3 and TcWRKY26 could form a protein complex in vivo. Our previous reports showed that transient *TcWRKY26* overexpression could remarkably increase *DBAT* expression. Yeast one-hybrid and luciferase activity assays revealed that TcWRKY26 could directly bind with the w_a_-box of the *DBAT* promoter to activate downstream reporter genes. All of these results indicated that TcWRKY26 acts as a direct regulator of *DBAT*, and the TcJAV3–TcWRKY26 complex is actually another JA signal transduction mode that effectively regulates taxol biosynthesis in *Taxus*. Our results revealed that JAV–WRKY complexes directly regulated *DBAT* gene in response to JA stimuli, providing a novel model for JA-regulated secondary metabolism. Moreover, JAV could also transduce JA signal and function non-redundantly with JAZ during the regulation of secondary metabolisms.

## 1. Introduction

Jasmonates (JAs), a kind of endogenous hormone that regulates plant growth and development, are considered one of the most effective inducers of secondary metabolite biosynthesis in various plants [1,2]. Moreover, as a hormone, JAs have been certified as a vital defense signal against herbivorous insects and necrotrophic pathogens [3]. Thus, the JA signaling transduction pathway has been deeply investigated in defense response [4], seed germination [5,6], seed size [7], regeneration [8], and others [9] over the past decades. Since JA is the most effective inducer of secondary metabolism, the JA-responding mechanisms of the biosynthesis genes of these secondary metabolites were deeply investigated [10,11,12].

JA regulates secondary metabolite biosynthesis dependent on jasmonate ZIM domain (JAZ) and its interactors, including MYCs, bHLHs, and MYBs [2]. For example, several investigations have elucidated a clear JA regulation mode of artemisinin biosynthesis in *Artemisia annua*. In brief, JA accumulation would promote the ubiquitination degradation of AaJAZs, resulting in the release of AaMYC2, which could directly activate *CYP71AV1/DBR2* by binding with G-box-like elements [13]. The JA regulation mode is conserved in different secondary metabolisms, such as NtJAZs–NtMYC2 in nicotine biosynthesis and CrJAZs–CrMYC2 in vinblastine biosynthesis [14]. All of these studies showed that JAZ repressors are crucial components in transducing JA signals to the biosynthesis genes of secondary metabolites. Taxol, also called paclitaxel, is produced from *Taxus* spp. and used as the most broad-spectrum anticancer drug [15]. Similar to other secondary metabolites, JA is able to greatly promote taxol biosynthesis. However, different from most other secondary metabolites, such as artemisinin, whose biosynthesis only needs four or five enzymes, taxol has a highly complicated chemical structure and needs more than 19 enzymes to catalyze the diterpene universal precursor, geranylgeranyl pyrophosphate, to the final product [16]. Among 14 known enzymes, taxane synthase (TASY) and 10-deacetylbaccatin III-10β-O-acetyltransferase (DBAT) were functioned crucially for taxol biosynthesis; TASY decides the flux because it is the first enzyme catalyzing universal diterpene precursor and produces crucial skeleton; DBAT acts as a rate-limiting enzyme [17,18]. Additionally, *TASY* and *DBAT* are early JA-responsive genes (Figure 1a) [19]. However, only *TASY* had been clearly elucidated as the JA-responding mechanism. A similar JA transduction pathway to *TASY* consisting of TcJAZ3, TcMYC2a, and the downstream factor TcERF15 was found. TcMYC2a and TcERF15 regulate *TASY* by directly binding with the E-box and GCC-box, respectively [18,20,21]. However, no solid evidence showed that the two factors were direct regulators of other genes. That is, the current JA transduction mode mediated by TcJAZs was not sufficient to explain the JA-responsive mechanism of all genes. For instance, the core promoter region of *DBAT* had no E-/GCC-/MBS-box, which codes a key rate-limiting enzyme, or no JAZ-MYB complexes were found to regulate taxol biosynthesis [19,22,23]. These results implied that other ways exist to transduce JA signals to taxol biosynthesis in addition to mediation by JAZs.

JAV1 might be the missing link of JA signals and *DBAT* gene. Hu et al. found that, independent of JAZs, AtJAV1 could also transduce JA signals and function similarly as JAZs to activate defensive gene expression and elevate resistances against insects and pathogens [24]. AtJAV1 belongs to VQ-motif-containing protein (VQP), which is a class of plant-specific proteins with a conserved and single short FxxhVQxhTG amino acid sequence motif [25]. Importantly, VQP, including AtJAV1, could physically interact with several WRKY transcription factors, which belong to Groups I and IIc [26]. Interestingly, current effective regulators of the *DBAT* gene were all WRKYs, including TcWRKY1 (Group IIa), TcWRKY26 (Group I), and TcWRKY33 (Group I) [23,27,28]. The results gave us a clue that the JA regulation of the *DBAT* gene might be mediated by JAV–WRKY complexes. Moreover, the two Group I WRKYs, TcWRKY26 and TcWRKY33, might be the direct downstream factor of TcJAVs. Previously, TcWRKY26 was found to be a MeJA-responsive WRKY factor, and its overexpression increased *DBAT* expression in *Taxus* suspension cells [27], whereas TcWRKY33 was a SA-responsive factor (Chen et al. 2021). Thus, TcWRKY26 was the most potential downstream factor of TcJAVs.

In this study, we aimed to screen out the homolog of AtJAV1 in *Taxus* and verify whether the JAV–WRKY mode exists and effectively regulates the *DBAT* gene. TcJAV3 was identified to be closest to AtJAV1 through homolog search from multi-omics data. The *TcJAV3* gene was complemented into the knockdown mutant *JAV1Ri17*, the *Arabidopsis* transgenic RNAi line of AtJAV1, to confirm the functions of TcJAV3 in JA signal transduction. We compared the disease resistance of complement lines, *JAV1Ri17* and wild types, to evaluate the functions of TcJAV3. Then the interactions of reported TcWRKYs with TcJAV3 were conducted by using a yeast two-hybrid system (Y2H). The *DBAT* promoter was studied in detail. The interacting TcWRKY was checked to determine whether it is a direct regulator of the *DBAT* gene. Our results revealed that JAV–WRKY complexes take part in defense and play important roles in secondary metabolisms. In addition, two JA transduction branches, JAV–WRKY and JAZ–MYC2, are equally important in transducing JA signals, although they might have different targets.

## 2. Results

### 2.1. TcJAV3, a Typical VQP, Was Closest to AtJAV1

VQP was named after a 10 aa conserved VQ-motif in their proteins. Apart from the short VQ motif, VQP has a relatively low sequence homology. Thus, 34 *Arabidopsis* VQPs were quite divergent and probably functioned in various bioactivities [26]. Now, AtVQ22, also called AtJAV1, was confirmed to function in the JA signaling transduction pathway [24]. Therefore, TcJAV3 was sequence aligned and phylogenetically analyzed.

The alignment showed that the 10 aa VQ-motif and its N-terminal 25 residues, a total of 34 aa (95–128 aa), were highly conserved between TcJAV3 and AtJAV1 (Figure 1b). However, the other parts of these proteins had barely any similarity (Appendix A). A comparison of TcJAV3 with other AtVQPs showed a similarity in 20 residues (about 109–128 aa), including the VQ-motif (Appendix A). The results suggest that TcJAV3 was more similar to AtJAV1, and the 95–109 aa might have an unknown but important function for TcJAV3 and AtJAV1. Consistent with previous studies, a majority of the branches had low confidence values, including the branching containing TcJAV3 and AtJAV1 (Figure 1c and Appendix A). Nonetheless, TcJAV3 was the closest to AtJAV1 and its homolog, indicating a similar function between them. 

### 2.2. TcJAV3 Complemented Arabidopsis Knockdown Mutant JAV1Ri17 

*TcJAV3* was complemented to *AtJAV1* knockdown *Arabidopsis* line, *JAV1Ri17*, under *CaMV 35S* promoter control for further study because of the presence of divergent sequences. In *JAV1Ri17*, *AtJAV1* is silenced; mutant plants have enhanced resistance to necrotrophic pathogens and herbivorous insects [24]. Our results were similar with the previous report; that is, the knockdown *JAV1Ri17* line was resistant to *B. cinerea* by phenotypic observations, including disease severity, lesion area, and plant-survival percentage. After *B. cinerea* spore suspension was sprayed for 5 days, *coi1-2* barely survived, and all leaves were severely disease infected. About 84% of *JAV1Ri17* lines survived; only 73% and 75% survived for the Col-0 and *TcJAV3* complement lines, respectively; and no remarkable differences were found between the WT and TcJAV3 complement lines (Figure 2a–c).

Similarly, in the drop inoculation experiments, *coi1-2* exhibited the most serious disease symptoms, with the largest lesion areas, but *JAVRi17* showed a highly strong resistance. WT and *TcJAV3* complement lines showed middle disease symptoms without difference (Figure 2d,e). The results demonstrate that the complementation of TcJAV3 indeed covered AtJAV1 silencing, and TcJAV3 performed the same function in the JA signal transduction pathway as AtJAV1. Thus, TcJAV3 was capable of transducing JA signals in *Taxus*. 

### 2.3. TcWRKY26, a Typical Group I WRKY Factor, Physically Interacted with TcJAV3

According to previous reports, VQP commonly interacted with WRKY factors but only the members of Groups I and IIc [26]. Thus, we screened WRKYs from these two subtypes according to our previous reports [27]. 

TcWRKY26 contained two WRKY domains, WRKY_NT and WRKY_CT, which are characteristic of Group I WRKY proteins, whereas Groups II and III had only one WRKY domain each (Figure 3a and Appendix A). The phylogenic analysis also showed that TcWRKY26 was clustered with known Group I WRKY factors with high values, and TcWRKY26 was the closest to AtWRKY20 (Figure 3b).

Only four amino acid residues were found between the two conserved Cys residues (Cx4C) in the zinc-finger structure of the C- and N-terminal WRKY domains of TcWRKY26 (Figure 3b and Appendix A). Additionally, two Asp residues, D491 and D494, which immediately precede the WRKYGQK signature sequence in the WRKY-CT of TcWRKY26, were the same with the D359 and D362 of AtWRKY33. Cx4C and two D residues are critical for interaction with VQPs [26]. 

Using Y2H, the yeast cells containing TcJAV3 and TcWRKY26 could grow in SD/-TLHA-deficient medium, indicating that TcJAV3 could bind with TcWRKY26 in yeast (Figure 3c). Moreover, TcJAV3 and TcWRKY26 were fused with the C- and N-terminus EYFP, respectively. Then fluorescence microscopy revealed a clear yellow fluorescence in tobacco leaves (Figure 3d). Luciferase complementary imaging (LCI) also verified that TcJAV3 could physically interact with TcWRKY26 in tobacco (Figure 3e). 

### 2.4. TcWRKY26 Was a Direct Regulator of DBAT by Binding with W-Box in Its Promoter

The overexpression of the transient cell lines of *TcWRKY26* could upregulate the *DBAT* gene by two folds [27]. However, we do not know whether TcWRKY26 directly regulates the *DBAT* gene. Many studies reported that two w-boxes are crucial cis-elements in the *DBAT* promoter [23,28] (Figure 4a).

Thus, we first tested the binding affinity of TcWRKY26 and the two w-boxes by using the yeast one-hybrid system (Y1H). The w_a_-box and w_b_-box were triplicated and ligated into pHIS2.1 to generate bait vectors. Then TcWRKY26-AD was co-transformed into yeast Y187 cells with two bait vectors (Figure 4b). The positive transformers containing TcWRKY26-AD with w_a_- or w_b_-box could grow in all mediums by selecting the deficient medium, and these transformers with the w-box were inhibited in SD/-TLA with more than 100 mM 3-AT (Figure 4c). Such results indicated that TcWRKY26 could bind with w_a_- or w_b_-box and that w_a_-box had a higher affinity than w_b_-box.

Subsequently, the original *DBAT* promoter and progressive deletion fragments were separately cloned to the 5′ terminus of luciferase to generate three reporters (Figure 4d). Co-transformation with the effector *35S*-TcWRKY26 showed that only complete *DBATp* had an increased LUC activity in tobacco leaves. The fragments containing w_b_-box only or no w-box did not show any difference when co-transformed with TcWRKY26. The result indicated that TcWRKY26 directly upregulated the *DBAT* gene by binding with w_a_-box.

## 3. Discussion

AtJAV1 belongs to VQPs and is characterized by a 10 aa peptide (labeled as FxxhVQxhTG). However, excluding the VQ-motif, no similarity exists between VQPs in the same species [25]. Thus, screening out the homolog of AtJAV1 in *Taxus* was crucial in our work. 

According to high-throughput data, TcJAV3 was found to be the most relative with AtJAV1 from total 24 full-length TcVQ proteins in *T. chinensis*. Only the residue F of the VQ-motif in TcJAV3 was different with the Y residue of the VQ-motif in AtJAV1 (Figure 1a). Moreover, an N-terminus with a 25 aa peptide, which immediately precedes the VQ-motif, was also highly conserved with AtJAV1, suggesting that TcJAV3 and AtJAV1 evolved more strictly. Moreover, TcJAV3 was grouped with AtJAV1 and its homologs but not other *Arabidopsis* VQPs, although the branch value was low. All of these results indicated that TcJAV3 is a potential AtJAV1 homolog in *Taxus*.

Therefore, we complemented *TcJAV3* into the *Arabidopsis* AtJAV1 knockdown line, *JAV1Ri17*, to test whether TcJAV3 could recover the phenotype change caused by AtJAV1 knockdown. *JAV1Ri17* exhibited a stronger resistance to pathogen infection, such as *B. cinerea* infection [24]. AtJAV1 functions as a negative regulator to control plant defense. Whole-plant spray inoculation and drop inoculation tests indicated that the complement transgenic lines of *TcJAV3* were more susceptible to *B. cinerea* infection than *JAV1Ri17* (Figure 2). Complement transgenic lines displayed similar disease symptoms with Col-0, indicating that TcJAV3 actually played similar roles as AtJAV1. Otherwise, TcJAV3 could transduce JA signals in *Taxus* with high possibilities. 

Similar to JAZ proteins, JAVs also recruit several transcription factors to form a complex. Moreover, JAV proteins were found to physically interact with WRKY factors [26]. Interestingly, we and co-workers had found that there were two w-boxes that specially bind with WRKY, localized in the key site of *DBAT* promoter region previously. Although these two cis-elements were first identified as salicylic acid (SA)-responsive elements, many JA-responsive WRKY factors bind with them, such as TcWRKY1 [23]. More WRKY factors were found to bind with the two w-boxes, suggesting that the two w-boxes are crucial for *DBAT* gene expression [28]. Interestingly, JAV1 transduces JA signals by recruiting WRKY factors. Such information notified us that JAV1 and some WRKYs might be the missing link for *DBAT* to respond to JA signals. 

Cheng et al. found that two structural features of WRKY domains are critical for interaction with VQPs. D359 and D362, which precede the WRKYGQK-motif in AtWRKY33 (Group I), are critical residues for interaction with VQPs. The other feature is that only four amino acid residues are present between the two conserved Cys residues (Cx4C) in the zinc-finger structure of the C- and N-terminal WRKY domains. An insertion mutant of AtWRKY33 lost the ability to bind with VQ10 [26]. 

Therefore, we screened these WRKYs from reported functional TcWRKY factors. Through Y2H screening, a Group I factor, TcWRKY26, was validated to interact with TcJAV3 in yeast. We also performed BiFC and LCI experiments to further confirm the interaction of TcJAV3 and TcWRKY26. Previously, TcWRKY26 was found to be a MeJA-responsive WRKY factor, and its overexpression increased *DBAT* expression in *Taxus* suspension cells [27]. However, no proof has shown that TcWRKY26 is a direct regulator of the *DBAT* gene, and how TcWRKY26 responds and transduces JA signals are not clear. W-boxes (w_a_- and w_b_-box) are the key cis-elements in the *DBAT* promoter [23]. Thus, TcWRKY26 could directly bind with these w-boxes and then regulate *DBAT* gene expression. In this study, we verified that TcWRKY26, indeed, directly binds with the two w-boxes, especially w_a_-box, to regulate *DBAT*, as determined by Y1H and LUC activity assays (Figure 4). All of these results concluded that the TcJAV3–TcWRKY26 protein complex is the key component to transduce the JA signals to the *DBAT* gene. Moreover, our results indicated that the JAV–WRKY complex plays important roles in the biosynthesis of secondary metabolites as important biotic defenses.

JAV1 is a freshly identified component of the JA signal transduction pathway; thus, not much is known about its functions in the regulation of secondary metabolite biosynthesis [24,29]. Hu et al., who first found JAV1, thought that JAV1 is a master controller that regulates JA-mediated plant defense but does not play a detectable role in plant development in *Arabidopsis*. However, according to our results, the JA signaling transduction pathway mediated by JAV also has multiple roles but not in plant defense. Indeed, components related to JAZ, including COI1, bHLH transcription factors (MYC2, MYC3, and MYC4), R2R3-MYB transcription factors (MYB21 and MYB24), and the WD-repeat/bHLH/MYB complexes, all play dual roles in regulating defense responses and diverse developmental processes [30,31,32,33]. Therefore, JAZ and JAV made JA a comprehensive and master hormone in regulating various plant bioactivities.

Current studies reported two ways to respond to JA signals for JAV. Ali et al. found that the ring-type E3 ubiquitin ligase JUL1 targets AtJAV1 and functions like COI1, which targets and mediates JAZ degradation [29]. The other way was that JAV–JAZ–WRKY forms a triple complex (JJW complex), and injury would cause the disintegration of the JJW complex by JAV1 phosphorylation, resulting in degradation [34]. Further studies were needed to clarify the detail of the response mechanism mediated by TcJAV3 (Figure 5). 

In conclusion, our work clarified another JA transduction pathway mediated by TcJAV3 and TcWRKY26 in *Taxus* that was different from the classical transduction system mediated by JAZ and its interactors (Figure 5). Moreover, the transducing complex consisting of TcJAV3 and TcWRKY26 was the key and direct reason for the response of *DBAT* to JA signals. The *DBAT* gene encodes the step-limited enzyme in taxol biosynthesis and was discovered to quickly respond to JA signals early. However, how *DBAT* responds to JA signals was unclear. According to our results, TcJAV3 and TcWRKY26 were the missing links between JA signals and the *DBAT* gene. These results completed the regulation mechanism of JA signals on taxol biosynthesis and are valuable for related research on the functions of JAV in the biosynthesis of various secondary metabolites.

## 4. Materials and Methods

### 4.1. Plant and Pathogen Materials

Three-year-old *Taxus media* plants were cottage propagated from one mother plant, which was bought from Muyang Xiangwen Landscaping Co., Ltd, Suqian, China. The *Arabidopsis thiana* wild-type Col-0 and *Nicotiana benthamiana* were preserved and bred in our lab. The knock-down mutant *JAVRi17* and T-DNA insertion mutant *coi1-2* were provided by Prof. Xie’s lab, Tsinghua University. Both *Taxus media* and *Arabidopsis thiana* grew in the greenhouses at 21 °C, with a 16 h/8 h light/dark photoperiod. Moreover, *Nicotiana benthamiana* grew at 25 °C with the same photoperiod. For the *A. thiana* used in whole-plant spray incubation, the plants were under a short-day period with 14 h/10 h light/dark (short day, SD) for 5 weeks. 

*Botrytis cinerea* 05.10 was preserved in PDB (Potato Dextorse Broth) medium by our lab.

### 4.2. Obtaining, Sequence Alignment, and Phylogenic Analysis of TcJAV3 and TcWRKY26

Fresh *Taxus* leaves were harvested and quickly frozen by liquid N_2_. After crushing, the total RNA was isolated by using HiPure HP Plant RNA Mini Kit (Magen). The first-strand cDNA templates were generated according to the protocol of TransScript^®^ One-Step gDNA Removal and cDNA Synthesis SuperMix (Transgene, Beijing, China). According to the previous report, primers were designed to amplify*TcWRKY26* [27]. Finally, we used PCR to amplify the two genes. 

To identify TcJAV3, several transcriptome data and *Taxus* genome data were obtained from public databases. We used local blast tools (NCBI blast+ v2.8) to search the homologues of AtJAV1, and then these fragments with the FxxhVQxhTG were further investigated and labeled TcJAVs. All of these TcJAVs and AtJAV1-32 were aligned by ClustalW with Protein weight matrix GONNET (https://npsa-prabi.ibcp.fr/cgi-bin/npsa_automat.pl?page=/NPSA/npsa_clustalw.html, accessed at 22 May 2022). 

The sequence alignment figures were all colored by Espript3.0 (https://espript.ibcp.fr/ESPript/cgi-bin/ESPript.cgi, accessed at 22 May 2022). Then the alignment files were submitted to MEGA-X for phylogenetic analysis to generate a neighbor-joining tree with 1000 resampling. 

### 4.3. Complement Vector Construction and Arabidopsis Transforming

The *TcJAV3* fragments were ligated into plant overexpression plasmid pBinGlyDsRED-35S with *Eco*R I and *Xho* I restriction sites [35], and then the complement vector 35S:TcJAV3OE was obtained. The alignment of *TcJAVA3* gene and *AtJAV1* gene are shown in Appendix A. Moreover, the plasmid was transformed into *Arabidopsis* (Col-0 ecotype) with *Agrobacterium tumefaciens* GV3101, using the floral dip method [36].

### 4.4. Botrytis Cinerea Growth and Plant Inoculation

*B. cinerea* was grown on PDB for 7–10 days at 20 °C under a 12 h photoperiod. Spores were collected and suspended in water containing 0.1% Triton X-100. In this study, both whole-plant spray incubation and drop incubation were used to evaluate the disease resistance, and *coi1-2* mutants were the positive control.

For drop incubation, the *Arabidopsis* leaves were plated on 1/2 MS with 1–1.5% agar, and then 5 µL of *B. cinerea* spore suspension (2.5 × 10^4^ spores/mL) was dropped onto the central part of leaves. Then these plates were incubated under high humidity at 22 °C with a 12 h photoperiod. Each genotype had 12 leaves at least. 

At 48 h post-inoculation, the lesion area (cm^2^) on each leaf was measured by using ImageJ (National Institutes of Health). All experiments were repeated three times. Error bars denote ±SEM. According to the one-way ANNOVA Duncan test, asterisks indicate statistically significant differences compared with WT (** *p* < 0.01). 

For whole-plant spray incubation, 5-week-old plants cultured under SD conditions were sprayed with 2.5 × 10^4^ spores/mL of *B. cinerea* spore suspension (~2 mL/plant). The plants sprayed with water were the negative control. Then these plants were grown under consistent humid conditions for 7 days; the first 2 days should be dark, but the remaining 5 days consisted of a normal photoperiod. The standard of disease severity was determined according to reference Hu et al. [24]. The survival plants were judged according to the disease severities of all leaves in each plant. The plants were thought to survive if less than 50% of the leaves had lesser disease severity (<50%). At least 12 plants from each genotype were used in each experiment, and all experiments were repeated three times. The survive ratio was statistical according to Student’s *t*-test. 

### 4.5. Yeast Two-Hybrid

TcJAV3 and TcWRKY26 were cloned into pGBKT7 via *Eco*R I/*Xho* I sites and pGADT7 via *Eco*R I/*Sal* I sites to generate TcJAV3-BD- and TcWRKY26-AD-fused protein, respectively. Particularly, the *TcJAV3* fragments were added with *Eco*R I and *Xho* I restriction sites, while pGBKT7 was linearized with *Eco*R I and *Sal* I. Since *Xho* I and S*al* I have the same cohesive end, the fragments and linear plasmid can be ligated by T4 ligase. The primers used are listed in Appendix A. 

The yeast AH109 competent cells were obtained by 1.1 × TE/LiAc treatment, and the AD and BD vectors were co-transformed into AH109 cells by PEG/LiAc method. Then the transformers were plated on solid selective medium (SD/-Trp/-Leu, SD-TL) and cultured at 30 °C for 2–4 days. Then the cultures of positive transformers were gradient diluted (originally OD600 = 2.0), and they were dropped on SD-TL, SD-Trp/-Leu/-His (SD-TLH), and SD-Trp/-Leu/-His/-Ade (SD-TLHA) plates. The plates were grown at 30 °C for 3–5 days and then observed. 

### 4.6. Bimolecular Fluorescence Complementation (BiFC)

TcJAV3 and TcWRKY26 were cloned into pSPYCE(M) via *Bam*H I/*Sal* I sites and pSPYNE(R)173 via *Bam*H I/*Xho* I sites to generate TcJAV3-CE- and NE-TcWRKY26-fused protein, respectively. TcJAV3-CE and NE-TcWRKY26 vectors were transformed into GV3101, respectively. The positive transformers were harvested, resuspended, mixed by 10 mM MgCl_2_ with10 mM MES (pH = 5.6) and 40 µM AS, and then stewed for 2 h. Then the mixtures were injected into leaf blade abaxially, and the injected plants were dark cultured for 12 h; they were then cultured under normal light conditions for 48 h. Finally, fluorescence was observed by Laser confocal microscope ZEISS 880 (ZEISS, Germany). The leaves containing TcJAV3-CE and empty pSPYNE(R)173 vector (NE), empty pSPYCE(M) (CE), and NE-TcWRKY26 were used as negative controls. 

### 4.7. Luciferase Complementary Imaging

TcJAV3 and TcWRKY26 were ligated into JW771N-LUC and JW771C-LUC to generate CLUC-TcJAV3- and TcWRKY26-NLUC-fused protein, respectively. The restriction enzymes were *Bam*H I and *Sal* I in each case. After *Agrobacterium* infiltration of indicated combinations in *N. benthamiana* leaves for 2 days, the luciferase substrate (luciferin) was sprayed onto the surface of the leaves, and luminescence was detected with an Imaging System (NEWTON 7.0Bio, Vilber, France).

### 4.8. Yeast One-Hybrid 

The two w-boxes, w_a_- and w_b_-box, were respectively ligated into pHis2.1 plasmid to generate two baits. The triplication fragments of w_a_- and w_b_-box were artificially synthesized. In order to ligate with linear plasmid, the sense and antisense sequence were added the cohesive end of *Eco*R I at 5′ end and *Spe* I at 3′ end, so that after annealing, two single chains would form a double chain with cohesive end of specific restriction site. Then the double chain could be ligated with the linear pHIS2.1, which was digested by *Eco*R I and *Spe* I. 

The method of preparation of yeast Y187 competent cells was the same as that of AH109. The co-transforming pH was 2.1–3 × Wa-box or pH is 2.1–3 × Wb-box with TcWRKY26-AD into Y187; the transforming and selecting were both the same as previously described. Finally, the positive transformers were plated on SD-TLH medium with different concentrations of 3-AT. 

### 4.9. LUC Activity Assay

*TcWRKY26* was cloned into pCAMBIA3302Y with *Eco*R I and *Sma* I, finally obtained the effector vector, pTcWRKY26-OE. For the reporter vectors, progressive deletion fragments of *DBAT* promoter were ligated into pGreenII0800-LUC and generated pGreenII0800LUC-DBAT, pGreenII0800LUC-DBAT-1, and pGreenII0800LUC-DBAT-de-Wbox. The reporter pGreenII0800LUC-DBAT-1 lacked w_a_-box, and pGreenII0800LUC-DBAT-de-Wbox lost the two boxes. 

Differently, effector vector pTcWRKY26-OE was transformed into GV3101, while three reporter vectors were transformed into GV3101 containing pSOUP plasmid. Then the effector and each reporter were co-transformed into *N. benthamiana* leaves, respectively, according to the previous method. After 48 h, LUC activities were determined by Dual Luciferase Reporter Gene Assay Kit (YEASEN, Shanghai, China). All experiments were repeated three times. The survival ratio was statistic according to the Student’s *t*-test.

## Figures and Tables

**Figure 1 ijms-23-13194-f001:**
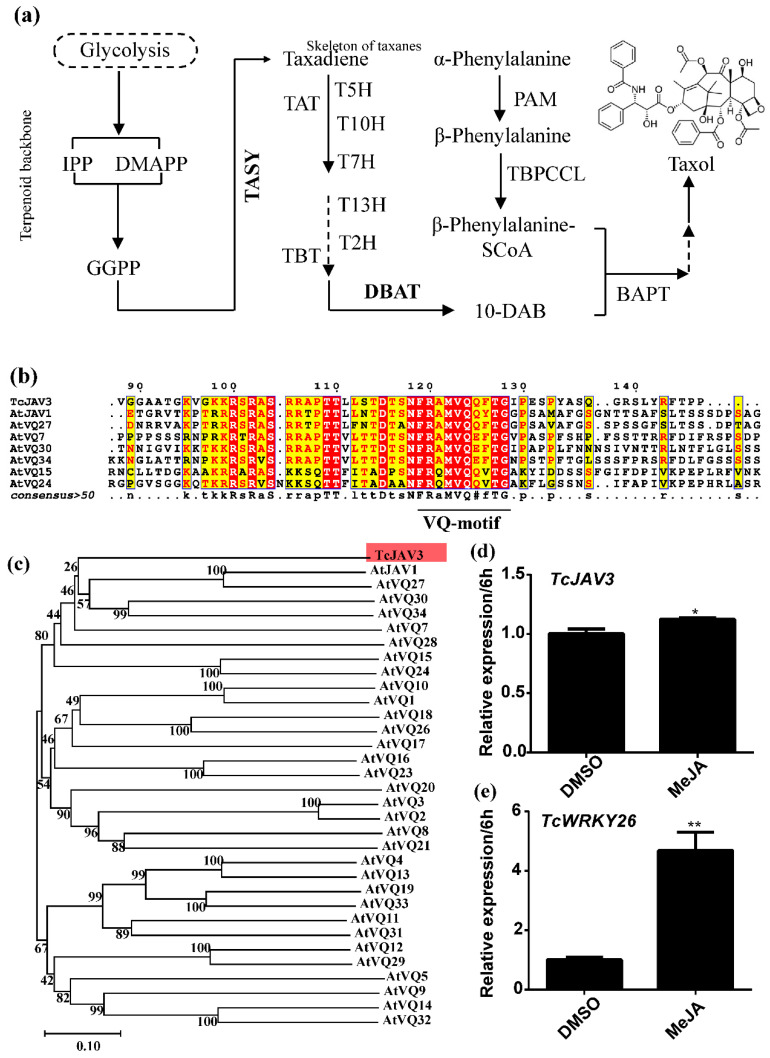
Characterization of TcJAV3 (**a**) Taxol biosynthesis pathway. GGPP (geranylgeranyl pyrophosphate) is the universal precursor of all diterpene, and TASY would cyclize it as the core structure of taxol. Then series of hydroxylation and acylation, including the 13′ side chain modification, take place on the core structure. DBAT is a rate-limiting enzyme in this pathway. (**b**) The aligned sequence of TcJAV3 and AtVQs. Besides the VQ-motif, the residues from 95 to 130 of TcJAV3 are highly similar with AtJAV1. (**c**) Phylogenetic tree of TcJAV3 and all AtVQPs. The gene expression levels of TcJAV3 and TcWRKY26 responding to JA are shown in (**d**,**e**). Statistics analysis was conducted as Student *t*-test; * represents *p*-value < 0.05, and ** represents *p*-value < 0.01.

**Figure 2 ijms-23-13194-f002:**
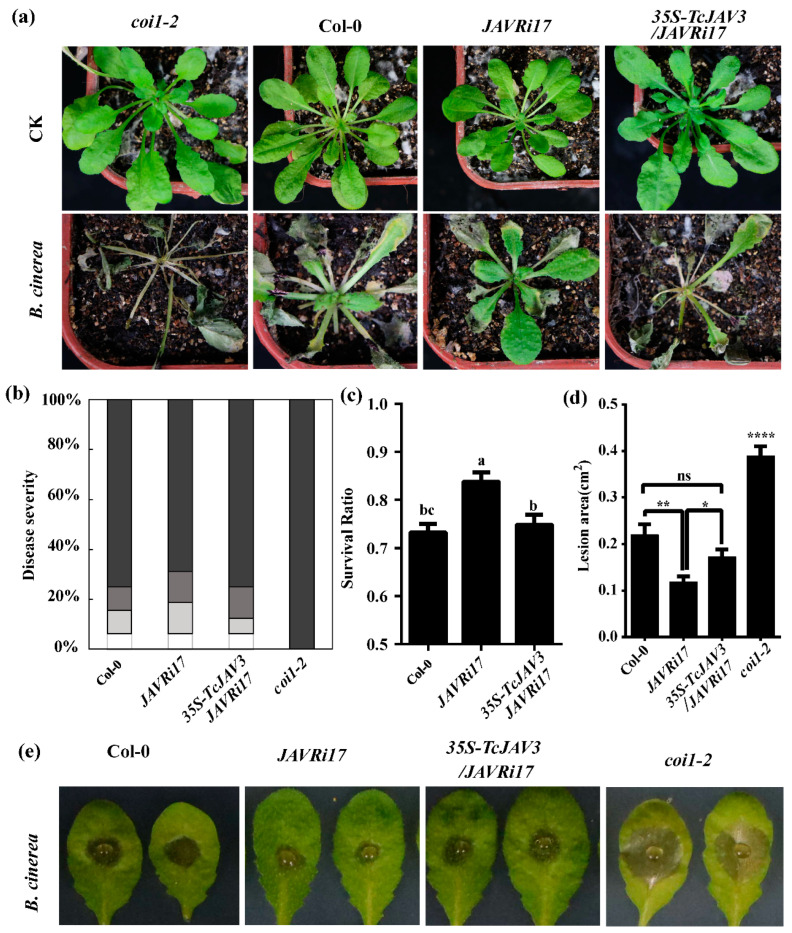
Disease-resistance phenotype in *TcJAV3* complement *Arabidopsis* line. (**a**) The phenotypes of *coi1-2*, WT (col-0), AtJAV1 RNAi transgenic plants (*JAVRi17*), and *TcJAV3* complement plants (*35S-TcJAV3/JAVRi17*) 7 days after spray inoculation with *B. cinerea* or water (CK). (**b**) Disease severity of the plants indicated in (**a**). The darker bars indicate the percentage of leaves with severe disease symptoms; the white or light gray bars indicate weak symptoms or no visible symptoms. (**c**) Survival percentage of the plants indicated in (**a**) (mean ± SEM; *n* = 12; statistics by one-way ANNOVA Duncan test). Bars with the different letter are significantly different from one another, *p* < 0.05. (**d**) Necrotic lesion area in each leaf described in (**e**) (mean ± SEM; *n* = 12; statistics by *t* test; * *p* < 0.01, ** *p* < 0.01, and **** *p* < 0.001). (**e**) The phenotypes of the leaves from WT, *coi1*, *JAVRi17*, and *35S-TcJAV3/JAVRi17* at 48 h after drop inoculation with 5 µL spore suspension of *B. cinerea* or with water (CK).

**Figure 3 ijms-23-13194-f003:**
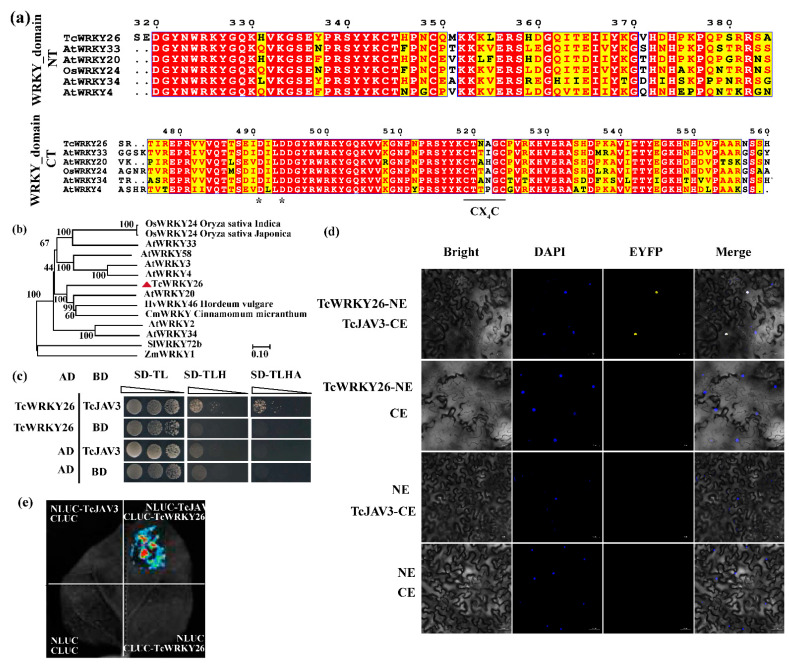
TcWRKY26, classical Group I WRKY, physically interacted with TcJAV3 (**a**) The two WRKY domains of TcWRKY26; only Group I WRKY factors have two WRKY domains. In the WRKY_domain_CT, the two D residues (labeled as *) and Cx_2_C motif are essential for the interaction with VQPs. (**b**) Phylogenetic tree of TcWRKY26 and reported Group I WRKYs. (**c**) Y2H results of TcWRKY26 and TcJAV3. SD/-TL: SD/-Trp-Leu; SD/-TLH: SD/-Trp-Leu-His; SD/-TLHA: SD/-Trp-Leu-His-Ade. (**d**) BiFC results of TcWRKY26 and TcJAV3. (**e**) LCI results of TcWRKY26 and TcJAV3. Bar = 50 μm.

**Figure 4 ijms-23-13194-f004:**
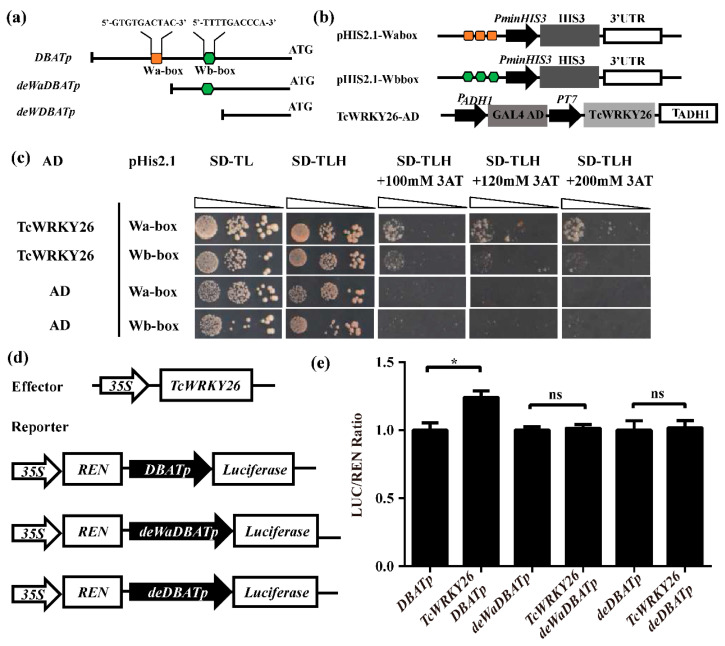
TcWRKY26 activates the *DBAT* gene by binding with w_a_-box. (**a**) Sketch map of progressive deletion fragments of *DBAT* promoter. (**b**) The vectors used in Y1H. Orange square and green hexagon indicate w_a_- and w_b_-box, respectively. The w_a_- and w_b_-box were tripled and ligated in the front of *HIS_minum_* promoter. (**c**) Y1H results of TcWRKY26 and w-boxes. SD/-TL: SD/-Trp-Leu; SD/-TLH: SD/-Trp-Leu-His. 3-AT: 3-amino-1, 2, 4-triazole. (**d**) The reporter and effector vector used in LUC activity assays. (**e**) LUC activity assays of progressive deletion of *DBAT* promoters in (**a**). Statistics analysis was conducted as Student *t*-test; * represents *p*-value < 0.05.

**Figure 5 ijms-23-13194-f005:**
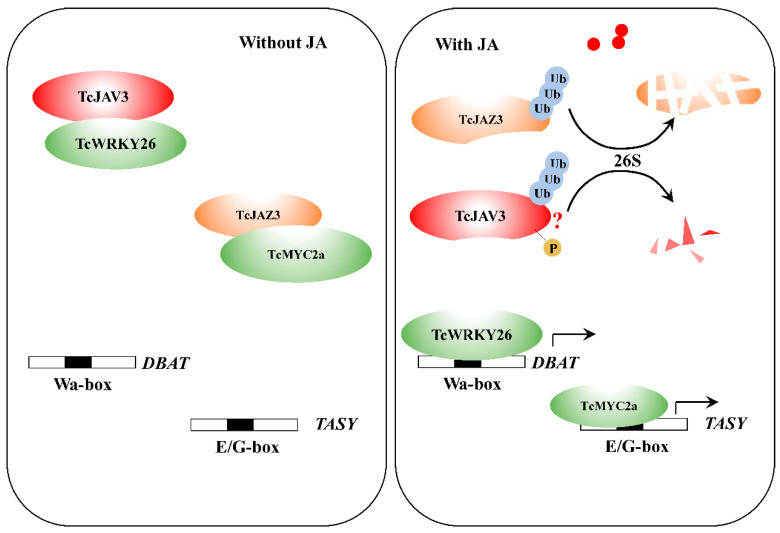
JA regulation mechanism of taxol biosynthesis. Without JA molecules, the TcWRKY26 and TcMYC2a were physically interacted with and inhibited by TcJAV3 and TcJAZ3, respectively. However, when plants suffered attacks of insects and pathogens, TcJAV3 and TcJAZ3 were ubiquitination degraded, TcWRKY26 and TcMYC2a were released to activate *DBAT* and *TASY* genes. However, according to current reports, there is another way: the triplet of JAZ–JAV–WRKY was destructed when JAV was phosphorylated, and then WRKY was released. Further studies were needed to clarify the details of the mechanisms.

## Data Availability

The sequence of *TcJAV3* was submitted to GenBank under accession number ON759763.

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
