# Peer review of "TcJAV3–TcWRKY26 Cascade Is a Missing Link in the Jasmonate-Activated Expression of Taxol Biosynthesis Gene DBAT in Taxus chinensis"

_ijms, 2022, doi:10.3390/ijms232113194_

Round 1

Reviewer 1 Report

The paper seeks to investigate the importance of JAV1 of Taxus and how it directly regulates the DBAT gene in response to JA stimuli. 

Overall I find it very good research, and well structured at the methodological level, but it has a major problem in demonstrating why TcJAV3 is used. In the alignment and phylogenetic tree, several arabidopsis thaliana sequences and only one taxus sequence are observed. When you want to demonstrate a homolog of a gene from another species by phylogeny, it is important to consider more than one sequence in the tree. Otherwise, the result is very forced. 

Fig1, all TcJAVs genes have to be added to the tree and in the alignment. 

In the introduction it does not explain to me why the use of TcWRKY26, it is only seen in line 274, I would also add the biological importance of WRKY26. 

Line 163, TcWRKY20 and HvWRKY46 appear, what biological relationship does it have with TcWRKY26? it is not clear why they appear in the text.

Homogenize nomenclature in gene naming, e.g., Fig3b, you use WRKY46 Hordeum vulgare, but the other genes are as AtWRKY. I recommend changing WRK46 to HvWRKY46 in the tree.

Line 230-237, to say this, I am still missing a comparison with the TcJAVs, which could be explained if you compare them at the sequence level. 

256-257 Missing reference

Author Response

Dear Reviewer: Thank you for your letter and for your comments concerning our manuscript entitled “TcJAV3–TcWRKY26 cascade is a missing link in the jasmonate-activated expression of taxol biosynthesis gene DBAT in Taxus chinensis” (ID: ijms-1932606). Those comments are all valuable and very helpful for revising and improving our paper, as well as the important guiding significance to our researches. We have studied comments carefully and have made correction which we hope meet with approval. The main corrections in the paper and the responds are as following: 1. Fig1, all TcJAVs genes have to be added to the tree and in the alignment. Response: Totally 32 VQ proteins in Taxus, but only 23 were full-length, according to your suggestion, all 23 TcVQs and 34 AtVQs were added to the tree and in the alignment. We made the alignment result to replace Figure S2, the tree was added as Figure S3. 2. In the introduction it does not explain to me why the use of TcWRKY26, it is only seen in line 274, I would also add the biological importance of WRKY26. Response: Thanks for your suggestion. We have added the reason why TcWRKY26 was chosen (line 102-106, page 4). Briefly, previous reports showed that TcWRKY26 was the only MeJA-responsive Group I factor among of all the reported functional WRKYs in Taxus. 3. Line 163, TcWRKY20 and HvWRKY46 appear, what biological relationship does it have with TcWRKY26? it is not clear why they appear in the text. Response: Thanks for your suggestion. We deleted the HvWRKY46, it’s actually not important here (line 180, page 9). 4. Homogenize nomenclature in gene naming, e.g., Fig3b, you use WRKY46 Hordeum vulgare, but the other genes are as AtWRKY. I recommend changing WRK46 to HvWRKY46 in the tree. Response: Thanks for your suggestion. The name had been changed in Fig 3b. 5. Line 230-237, to say this, I am still missing a comparison with the TcJAVs, which could be explained if you compare them at the sequence level. Response: Thanks for your suggestion. According to Reviewer 2’s suggestion, this paragraph had been deleted. But as you said in Q1, we added related results. 6. 256-257 Missing reference. Response: Thanks for your suggestion. According to Reviewer 2’s suggestion, this paragraph had been deleted.

Reviewer 2 Report

This manuscript entitled “TcJAV3–TcWRKY26 cascade is a missing link in the jasmonate-activated expression of taxol biosynthesis gene DBAT in Taxus chinensis” by Chen et al interpreted TcJAV3 function in recovering Arabidopsis JAV1 RNAi plant resistance to the pathogen. And authors verified that TcJAV3 was able to interact with TcWRKY26, which directly binds DBAT promoter to activate its expression. this story is interesting, but some concerns need to be explored as follows:

1.     Authors considered that TcJAV3-TcWRKY26 is a cascade of JA signaling. Thus, the JA and JA signal-genes should be employed to test whether they affected this cascade including expression changes.

2.     The title includes taxol biosynthesis, so the related genes and taxol content need to be checked in various genetic materials.

3.     In overexpression Arabidopsis, the TcJAV3 mRNA could be silenced by overexpression dsRNA of AtJAV1 in JAVRi17 plant cells. Thus, authors need to interpret the two gene nucleotide differences.

4.     Line 155-175, authors need to verify whether the TcJAV3 interacts with other TcWRKYs in Group I and II to strengthen WRKY26 function.

5.     The panel in Figure 3d is unclear.

6.     In this manuscript, the function of TcJAV3 is not shown excluding recovery of AtJAV1 knockdown resistance to the pathogen. It is better to improve this study through adding the taxol biosynthesis-related experiments. Additionally, the DBAT function need to be verified in taxol biosynthesis.

7.     The Discussion section needs to be revised, for example, the first two paragraphs are repeated with Introduction.

Author Response

Dear Reviewer:

Thank you for your letter and for your comments concerning our manuscript entitled “TcJAV3–TcWRKY26 cascade is a missing link in the jasmonate-activated expression of taxol biosynthesis gene DBAT in Taxus chinensis” (ID: ijms-1932606). Those comments are all valuable and very helpful for revising and improving our paper, as well as the important guiding significance to our researches. We have studied comments carefully and have made correction which we hope meet with approval. The main corrections in the paper and the responds are as following:

  1. Authors considered that TcJAV3-TcWRKY26 is a cascade of JA signaling. Thus, the JA and JA signal-genes should be employed to test whether they affected this cascade including expression changes.

Response: A very good concern. Previously, the expression changes of TcJAV3 and TcWRKY26 were tested, and it showed that TcWRKY26 was a JA-responsive factor (Zhang et al., 2018a; doi:10.1038/s41598-018-23558-1), while TcJAV3 was slightly induced by JA, and these results had been added into Figure 1 (Figure 1d&e). Interestingly, how JA affected such JAV-WRKY complex are still unclear yet (Ali et al. 2019 and Yan et al. 2018), so more effects should be further investigated.

  1. The title includes taxol biosynthesis, so the related genes and taxol content need to be checked in various genetic materials.

Response: It’s a very good suggestion. Firstly, TcWRKY26 was previously verified to up-regulated taxol biosynthesis in transient Taxus cells (Zhang et al., 2018). Actually, it’s valuable if the genetic materials of TcJAV3 or TcJAV3 TcWRKY26 knockout mutants were obtained and tested about their taxol content. But Taxus is a complicated gymnosperm tree, available methods to obtain genetic materials are hard, and transient materials are limited to clarify detailed molecular mechanisms. Therefore, in this study, we investigated the functions of TcJAV3 in Arabidopsis, and such results verified and emphasized the ability to transduce JA signals of TcJAV3 with TcWRKY26. Then TcWRKY26 could regulate taxol biosynthesis genes. But we are very grateful for your suggestions, the direct relationship between TcJAV3 and taxol biosynthesis will be further verified.

  1. In overexpression Arabidopsis, the TcJAV3 mRNA could be silenced by overexpression dsRNA of AtJAV1 in JAVRi17 plant cells. Thus, authors need to interpret the two gene nucleotide differences.

Response: It’s really a good concern that we ignored. According to your suggestion, we aligned the two genes, but there was no any similarity between two nucleotide sequences. Thus, TcJAV3 mRNA might not be silenced by overexpression dsRNA of AtJAV1 in JAVRi17 plant cells.

  1. Line 155-175, authors need to verify whether the TcJAV3 interacts with other TcWRKYs in Group I and II to strengthen WRKY26 function.

Response: Thanks for your suggestion. Of course, TcJAV3 could bind with other TcWRKYs belong to Group I and IIc, TcWRKY26 would not be the sole one. But, TcWRKY26 was the only reported one of Group I that responds to JA and was capable of upregulating taxol biosynthesis (such description was added in Introduction part, line 95-101, page 4). Therefore, TcWRKY26 was the most potential one to regulate taxol biosynthesis after transducing JA signals.

  1. The panel in Figure 3d is unclear.

Response: Thanks for your suggestion. We had modified the whole Figure 3, hope it’s fine now.

  1. In this manuscript, the function of TcJAV3 is not shown excluding recovery of AtJAV1 knockdown resistance to the pathogen. It is better to improve this study through adding the taxol biosynthesis-related experiments. Additionally, the DBAT function need to be verified in taxol biosynthesis.

Response: It’s indeed a good suggestion. As we answered in Q2, it’s better if we can obtain the results of taxol content affected by TcJAV3 in various genetic materials. But currently, it’s hard to get solid and useful proof. A transient cell line was not sufficient to prove the effects of taxol biosynthesis by TcJAV3. We look forward to follow your suggestions to clarify the direct relationship between TcJAV3 and taxol biosynthesis further.

  1. The Discussion section needs to be revised, for example, the first two paragraphs are repeated with Introduction.

Response: Thanks for your suggestion. We deleted the two paragraphs in Discussion, and modify the Introduction part according to the two paragraphs. And we modified the whole Discussion section to make it simple but informative. 

Round 2

Reviewer 2 Report

alignment of TcJAVA3 gene and AtJAV1 gene required to be shown in supplemental files to support conclusion.

Author Response

Dear Reviewer:

Thank you for your letter and for your comments concerning our manuscript entitled “TcJAV3–TcWRKY26 cascade is a missing link in the jasmonate-activated expression of taxol biosynthesis gene DBAT in Taxus chinensis” (ID: ijms-1932606). We have studied comments carefully and have made correction which we hope meet with approval. The main corrections in the paper and the responds are as following:

Comments: alignment of TcJAVA3 gene and AtJAV1 gene required to be shown in supplemental files to support conclusion.

Response: Thanks for your suggestion, such result was added as Figure S5.
